# Verification of Outer Hair Cell Motor Protein, Prestin, as a Serological Biomarker for Mouse Cochlear Damage

**DOI:** 10.3390/ijms25137285

**Published:** 2024-07-02

**Authors:** Jing Zheng, Yingjie Zhou, Robert J. Fuentes, Xiaodong Tan

**Affiliations:** 1Department of Otolaryngology-Head and Neck Surgery, Feinberg School of Medicine, Chicago Campus, Northwestern University, Chicago, IL 60611, USA; robert.fuentes@northwestern.edu (R.J.F.); xiaodong.tan@northwestern.edu (X.T.); 2Department of Communication Sciences and Disorders, School of Communication, Evanston Campus, Northwestern University, Evanston, IL 60208, USA; yingjie.zhou@northwestern.edu; 3The Knowles Hearing Center, Northwestern University, Evanston, IL 60208, USA

**Keywords:** prestin, biomarker, cochlear damage, outer hair cell stress, HPβCD

## Abstract

The motor protein prestin, found in the inner ear’s outer hair cells (OHCs), is responsible for high sensitivity and sharp frequency selectivity in mammalian hearing. Some studies have suggested that prestin could be a serological biomarker for cochlear damage, as OHCs are highly vulnerable to damage from various sources. However, the reported data are inconsistent and lack appropriate negative controls. To investigate whether prestin can be used as a serological biomarker for cochlear damage or stress, we measured prestin quantities in the bloodstreams of mice using ELISA kits from different companies. Wildtype (WT) mice were exposed to different ototoxic treatments, including noise exposure and ototoxic reagents that rapidly kill OHCs. Prestin-knockout (KO) mice were used as a negative control. Our data show that some ELISA kits were not able to detect prestin specifically. The ELISA kit that could detect the prestin protein from cochlear homogenates failed to detect prestin in the bloodstream, despite there being significant damage to OHCs in the cochleae. Furthermore, the optical densities of the serum samples, which correlate to prestin quantities, were significantly influenced by hemolysis in the samples. In conclusion, Prestin from OHCs is not a sensitive and reliable serological biomarker for detecting cochlear damage in mice using ELISA.

## 1. Introduction

Hearing loss is a debilitating, life-altering disorder. Noise, age, and ototoxic drugs such as antibiotics and cancer drugs put stress on the cochlea, leading to hearing loss. According to the World Health Organization (WHO), by 2050, over 700 million people worldwide will be affected by hearing loss [1]. Unfortunately, no effective and reliable method exists for monitoring and detecting cochlear stress. Most importantly, it is impossible to identify cochlear stress before irreversible tissue damage and hearing loss have already occurred. Therefore, exploring the potential of cochlear stress biomarkers is an area of interest for researchers and clinicians. It may provide valuable insights into the pathology of cochlear damage and inform the development of more effective diagnostic and therapeutic interventions.

Mammalian hearing relies on the electromotility of outer hair cells (OHCs) to achieve high sensitivity and sharp frequency selectivity. In 2000, we identified prestin, a member of the anion transport family called SLC26A5, as the molecular basis for OHCs’ electromotility [2]. By changing its conformation between short and extended states, prestin subserves OHCs’ motility when switching between depolarized and hyperpolarized conditions [3,4,5,6,7,8]. Several lines of evidence have shown that the lateral membrane of OHCs is primarily occupied by prestin. OHCs from prestin-knockout (KO) mice lose their somatic electromotility, resulting in 40–50 dB hearing loss in prestin-KO mice [9,10,11]. OHCs without prestin proteins are also 40% shorter than the WT-OHCs [12]. Since OHCs are highly vulnerable to various insults, OHC-specific proteins are considered an excellent choice for revealing biomarkers that are useful in detecting cochlear stress and damage. By using a sandwich enzyme-linked immunosorbent assay (ELISA), prestin was detected in the bloodstream of humans [13,14,15], rats [16,17], guinea pigs [18,19], and mice [20]. Several reports have suggested that prestin in the bloodstream could be used as a biomarker for hearing loss such as idiopathic sudden sensorineural hearing loss [13,14,15,21,22], noise-induced hearing loss [16,17,23,24,25], sensory hearing loss [26,27,28], age-related hearing loss [29], ototoxic regents-induced hearing loss, such as HPβCD [30] and cisplatin [18,19,20,31], and also hearing loss observed in various diseases like Meniere’s Disease and Vestibular Migraine [32], COVID-19 [33], lead poison [34], and even surgery-related damage [21]. However, the data reported in several studies lack proper negative controls and show inconsistency [13,14]. To investigate whether prestin is a serological biomarker for cochlear damage or stress, cochlear homogenates from WT mice and prestin-KO mice [9] were used as positive and negative controls, respectively. These mice were also exposed to various ototoxic stimulations to cause cochlear stress, including noise exposure and ototoxic chemical treatment. HPβCD rapidly kills OHCs [35,36] and releases prestin into extracellular space. The serums and cochleae from WT and prestin-KO mice were collected at different time points after HPβCD injection or noise exposure. Prestin concentrations in the bloodstream and cochleae were measured using mouse prestin-ELISA kits purchased from three different companies, and the cochleae were also used for anatomic analysis. Our collected data suggest that prestin from OHCs is not a sensitive and reliable serological biomarker for detecting cochlear damage using ELISA.

## 2. Results

### 2.1. Confirm the Positive and Negative Control Samples for Prestin-ELISA: Prestin Is Expressed in OHCs from WT but Not from Prestin-KO Mice

The prestin-KO mouse model was created by removing the 3–7 exons of prestin, which encode the 245 N-terminal amino acids of prestin, including the ATG start code. Multiple laboratories, including ours, have studied the prestin-KO mice and confirmed that prestin is responsible for OHCs’ electromotility [9,12]. Prestin has 744 amino acids in total. To ensure that no prestin fragment was synthesized in the OHCs from prestin-KO mice, we performed immunofluorescence using antibodies against both the N- and C-terminals of prestin. Each anti-prestin antibody was tested in at least two prestin-KO mice along with cochlear samples from WT mice (WT n = 2, KO n = 2). As shown in Figure 1, prestin was detected in OHCs derived from WT mice, and not in OHCs from prestin-KO mice using anti-N-mprestin. Similar staining patterns were also observed when using anti-C-mprestin. The immunofluorescence data, using antibodies against both the N-terminal and C-terminal of prestin, confirmed that prestin was expressed in the WT mice but not in OHCs from prestin-KO mice. Thus, cochlear homogenates from prestin-KO and WT mice cochleae were used as negative and positive controls to verify prestin-ELISA kits’ specificity.

### 2.2. Test Sensitivities and Specificity of Different Prestin-ELISA Kits

We purchased prestin-ELISA kits from LSBio, Abbexa, and MyBioSource (MBS). All of these kits are sandwich ELISA kits. As shown in Figure 2A, prestin is captured and detected by two layers of anti-prestin antibodies (capture and detection antibodies) that bind to different prestin sites. Sandwich ELISA is known to have higher specificity and sensitivity than original ELISA as it involves using two antibodies with different epitopes. The detection ranges of these three kits are 78–5000 pg/mL for LSbio and Abbexa, and 7.8–500 pg/mL or 15.6–1000 pg/mL for MBS.

We collected cochleae around P16 (16 days postnatal) of the WT and prestin-KO mice to make cochlear homogenates because prestin-KO mice start to lose OHCs around P21 [12]. Equal amounts of cochlear homogenate from pooled WT and prestin-KO cochleae were used for ELISA using the kits purchased from the three different companies. The sample’s optical density (OD_450–570_), which correlates to its prestin levels, was measured along with different amounts of prestin proteins to make a standard curve. If the OD_450–570_ of any of the samples fell in the detection range, the OD_450–570_ numeric was converted to a prestin concentration based on the standard curves generated along with each measurement. As shown in Figure 2B, the results obtained from these three kits were quite different despite all of them detecting significantly different prestin levels between the WT and prestin-KO cochlear homogenates. For the LSBio kit, significant amounts of prestin proteins were detected in prestin-KO cochlear homogenate samples despite no prestin being present in the prestin-KO samples. The average prestin concentrations for the prestin-KO and WT samples were 681 pg/mL and 1034 pg/mL, respectively. These data suggest that the LSBio kit detects a significant amount of non-specific signals. For the Abbexa kit, all OD_450–570_ numeric values from the samples were below the detectable range (78 pg/mL) regardless of whether the cochlear homogenates were from the WT or prestin-KO mice, suggesting that the Abbexa kit is less sensitive than the others. For the MBS kit, the OD_450–570_ values of the prestin-KO samples were above the lowest limitation (7.8 pg/mL), suggesting non-specific signals’ detection by the MBS kit, similar to that observed in the LSBio kit case. However, the OD_450–570_ values of the WT samples were over the highest detectable range (500 pg/mL), seven times more than those of the prestin-KO samples. Because the OD_450–570_ numeric values obtained from the Abbexa and MBS kits fell outside the standard curves’ detectable range, the prestin quantities were expressed as OD_450–570_ instead of the protein concentration like those in the LSBio kit case. As shown in Figure 2B, the MBS kit has a better signal-to-non-specific noise ratio than the other two prestin-ELISA kits. Collectively, our ELISA data showed the variation in the performance of different prestin-ELISA kits in detecting prestin proteins, indicating the importance of verifying the specificity of prestin-ELISA kits.

### 2.3. Establish an OHC Damage Mouse Model to Test Whether Prestin from Cochleae Can Be Detected in the Bloodstream

The ability of the prestin protein to pass the ‘blood-labyrinth barrier’ and ‘leak’ into the vascular system is a crucial aspect of its potential as a serological biomarker for cochlear stress or damage. Prestin, a glycosylated membrane-bound protein with a molecular weight of 81.4 KD, forms dimers or tetramers embedded in the plasma membrane of the OHCs [37]. It remains unknown (1) whether extracellular prestin proteins released from the OHCs in the cochlea could pass the ‘blood-labyrinth barrier’ and enter the bloodstream, and (2) how long these native prestin proteins from OHCs can present in the bloodstream before they are degraded or excluded from the body. To answer these questions, we injected WT mice with HPβCD to release prestin from their OHCs. HPβCD is a cholesterol chelator that extracts cholesterol from plasma membranes. Because the plasma membranes of OHCs are enriched with prestin-bound cholesterol [35], we and others have discovered that HPβCD can rapidly rupture the OHC plasma membrane [35,36,38,39,40]. Thus, WT mice were injected with 8000 mg/kg of HPβCD or an equal amount of 0.9% NaCl solvent, as we described before [35,36]. We collected the cochleae from mice that were injected with either HPβCD or control saline within 3–4 h (4 h), 1 day (D1), 4 days (D4), 7 days (D7), and 11 days (D11), respectively. We performed immunostaining of the full length of the collected cochleae. Because artifacts caused by fixation, dissection, and immunostaining were more severe at the base and apex, we counted the OHC loss in the regions involving segment 3, which corresponds to frequency ranges of 19.1–36.5 kHz [41]. As expected, there was minimal OHC loss in the mice injected with 0.9% NaCl, regardless of the time when the cochleae were collected, ranging from D1 to D11 (n = 6) (Figure 3B). As shown in Figure 3, OHCs from 0.9% NaCl-injected mice showed green staining circles with uniform intensity, indicating that prestin was evenly distributed in the lateral membrane of OHCs. In contrast, within 3–4 h of HPβCD injection, OHCs exhibited uneven green circles with aggregate clots in the cytoplasm, indicating OHC deterioration but no significant OHC loss at this stage. As described in our previous study [35,36], significant OHC loss was observed just one day after the WT mice were injected with HPβCD (n = 3, *p* < 0.0001). From day four, no OHCs were observed in the middle regions of the OHCs, and some IHC and pillar cell loss was also observed (n = 2, *p* < 0.0001). These collected data confirm that a single high-dose administration of HPβCD resulted in mass OHC death in WT mice, but OHC loss was not observed in the WT control mice treated with saline. Thus, the HPβCD-treated mice were used as a model of damaged OHC/cochlea, as there were sufficient ‘released’ prestin proteins in the cochlea of the HPβCD-treated mice that were ready to pass the ‘blood-labyrinth barrier’ and ‘leak’ into the bloodstream.

### 2.4. Prestin Levels Show No Undetectable Difference in the Bloodstream of WT and Prestin-KO Mice Regardless of Whether OHCs Were Stressed or Damaged

To test whether prestin released from damaged OHCs can be detected in the bloodstream, both WT and prestin-KO mice were injected with HPβCD or control solvent, 0.9% NaCl, with prestin-KO mice serving as a negative control for the WT. Cochlear samples were collected and immunostained with phalloidin and Myosin 7A, which label inner hair cells (IHCs) and OHCs. The NaCl-injected mice had minimal OHC loss, as demonstrated in Figure 3, while the WT mice injected with HPβCD exhibited mass OHC loss (Figure 4A,C, n = 3), exceeding that in the prestin-KO mice injected with HPβCD (Figure 4B,D, n = 2). This observation is consistent with our previous data, confirming the prestin-dependency of OHCs’ susceptibility to the cholesterol chelator HPβCD [35]. Despite significant OHC loss or damage, the prestin concentrations in the serums from the HPβCD-treated WT mice did not differ significantly from those of the prestin-KO mice (no prestin), or the WT mice treated with NaCl control (no OHC loss) or no treatment at all. As shown in Figure 4, the Abbexa kit has a lower sensitivity than the other kits, and the OD_450–570_ values of all samples were lower than the detection limitation. The average prestin concentrations, measured by LSBio, for HPβCD-treated WT and prestin-KO, were similar: 2306 pg/mL for the WT and 2200 pg/mL for the prestin-KO samples, respectively. Since the MBS kit is the best one available, this kit was used to measure prestin concentrations in the serums collected from different conditions, including no treatment, one day, and seven days after the NaCl- or HPβCD-treatment of WT and prestin-KO mice. We tested two kits with different detection ranges. For the MBS kit with a detection range of 15.6–1000 pg/mL, we could detect prestin in cochlear homogenates but not in serum samples. As shown in Appendix A, the prestin signal was found in the cochlear homogenate (a positive control) but was not detected in the blood samples from both the WT and prestin-KO mice, with a few exceptions. Most of the numeric values for the OD_450–540_ were below the detectable limit of the prestin-ELISA kit, irrespective of whether the WT and prestin-KO mice were treated with HPβCD or NaCl. The serum samples with OD_450–540_ values that exceeded the detection limit were those with severe hemolysis (marked with *). For the MBS kit with a detection range of 7.8–500 pg/mL, as shown in Appendix A, the OD_450–570_ numeric values for the serum samples fell in the detection range. However, no significant difference between the WT and prestin-KO mice was found in all conditions, regardless of their treatments or the time at which the samples were collected. Considering the matrix effects often observed in serum samples in ELISA, we also diluted the serum samples in a 1:10 ratio rather than a 1:5 ratio with sample dilutant to increase the signal-to-noise ratio. However, we still did not find a significant difference between WT and prestin-KO mice regardless of whether they were treated with HPβCD, NaCl, or no treatment as shown in Figure 4G. We used multiple MBS kits generated with three different raw materials (different antibodies). Although the absolute OD_450–570_ numeric values were different for each ELISA, the general patterns remained the same; no significant differences in the OD_450–570_ numeric values between WT and prestin-KO were found. These collected data led us to doubt whether prestin can pass the ‘blood-labyrinth barrier’ and ‘leak’ into the vascular system.

To investigate the possibility of prestin leakage into the bloodstream and subsequent degradation before ELISA measurement, we followed a rigorous timeline to collect serum samples from WT mice injected with HPβCD: (1) 4 h group, 3–4 h after injection when the OHCs were under stress but not yet dead (Figure 3), (2) 24 h group, 1 day after injection when there was a significant loss of OHCs but some were still present (Figure 3), and (3) 4–7 days group, 4–7 days after injection when there were no OHCs left (Figure 3). To ensure consistency, littermates were used for HPβCD or NaCl injection in the same group. We measured prestin concentrations in the serums using the MBS kit, which has the best signal-to-noise ratio (Figure 2). As illustrated in Appendix A, there was no significant difference in the prestin concentrations in the serums regardless of the type or timing of the injection. The levels of prestin in the WT serums did not indicate the levels of OHC stress (HPβCD treated for 3–4 h), OHC loss (HPβCD treated for from 24 h to 7 days), or unstressed OHCs (NaCl-treated). Similar results were also found when more mice from different litters were added, as shown in Figure 5. Collectively, our data showed that prestin is not a sensitive and reliable serological biomarker for mouse OHC damage using ELISA.

### 2.5. The Severities of Hemolysis Influence Prestin Quantification Measured by ELISA

Several studies have shown that prestin concentrations in the blood of humans, rats, and guinea pigs change significantly after exposure to loud noise [16,17,23,24,25]. In our study, we conducted similar experiments in WT mice. WT mice were exposed to bandlimited (8–16 kHz) noise for 2 h at a 110 dB sound pressure level (SPL), which led to permanent hearing loss (PTS). Serum samples were collected to measure prestin concentrations using the MBS kit. In contrast to other studies, no significant differences in prestin concentrations in the serums were found before and after noise exposure (data not shown). However, we discovered that mouse red blood cells are fragile and can easily rupture under physical pressure. To test whether hemolysis may affect ELISA results, we collected two tubes of serums from the same mouse—one with hemolysis and one without. The hemolysis serums were red/pink, and the hemoglobin levels measured by absorbance at 414 nm were >0.5. An elevated OD_414_ is correlated with increased free oxyhemoglobin from hemolysis. Figure 6 showed three examples: WT mice exposed to loud noise, WT mice injected with HPβCD, or a negative control mouse, prestin-KO. As shown in Figure 6, serums with hemolysis have two–three times more OD readings than those without hemolysis (absorbance at 414 nm was 0.2), despite the fact that two samples were collected from the same animal at the same time. In other words, increased hemolysis elevated the measured prestin concentrations even for the negative control serums derived from prestin-KO mice, suggesting that the prestin-ELISA can detect non-specific proteins released from red cells unrelated to prestin. These data indicate that the severity of hemolysis significantly increases the non-specifical binding of prestin antibodies and influences the prestin concentrations measured by the prestin-ELISA.

## 3. Discussion

In this study, we investigated whether OHCs’ motor protein, prestin, could be used as a serum biomarker for cochlear damage. We addressed this question through a four-step experimental design. First, we confirmed that prestin is not expressed in OHCs from prestin-KO mice (Figure 1). Therefore, prestin-KO mice were used as a negative control. Second, we verified the specificity of three prestin-ELISA kits using cochlear homogenates derived from WT and prestin-KO mice (Figure 2). The specificity and sensitivity vary among these kits. Even the best kit, the MBS kit, has some false positive issues. Third, we tested whether prestin could be detected in serums after cochlear stress/damage (Figure 3, Figure 4 and Figure 5). All kits failed to specifically detect prestin signals in the serum in comparison to serums from prestin-KO mice. The prestin levels in the serum did not significantly change regardless of whether the OHCs were under stress, unstressed, or dead. Finally, we found that prestin concentrations measured by these ELISA kits are significantly affected by the quality of the collected serum (Figure 6). Our data analysis leads us to conclude that prestin is not a sensitive and reliable serological biomarker for mouse OHC damage using ELISA.

As we stated in the Introduction, numerous papers have utilized prestin as a biomarker for cochlear damage or stress through ELISA. However, none of these studies have employed a suitable negative control to validate the specificity of their ELISA. As the pioneering group in the discovery of prestin, we and other research teams have developed dozens of anti-prestin antibodies targeting different amino acids of prestin. Despite our extensive efforts, we have not yet identified an antibody that exclusively detects prestin. For instance, an antibody can specifically identify prestin expression in the cochlea by immunofluorescence, but it does not consistently do so in Western blot analysis (data not shown). This underscores the critical role of a validated ELISA in detecting authentic prestin in the serum. Our research is the first to utilize prestin-KO as the negative control, a gold standard for confirming the specificity of ELISA. Our data showed that all commercial ELISA tests failed to detect prestin specifically in the serum.

Based on our data, we cannot determine whether prestin proteins pass the ‘blood-labyrinth barrier’ and ‘leak’ into the vascular system because all prestin-ELISA kits that we tested failed to detect prestin specifically in the serum samples. The Abbexa kit was unable to detect prestin at all. Although the other two could detect prestin in cochlear homogenate, they also detect significant non-specific signals. The situation worsened when serum samples showed hemolysis. Prestin has 744 aa (81.4 KD) with 14 transmembrane domains. It is buried in the lipid membrane as a dimer or a tetramer, weighing 160 KD or 320 KD. These oligomers are not a minor subject. The ability of prestin to pass through the ‘blood-labyrinth barrier’ is essential for its use as a serological biomarker for cochlear stress. Therefore, we conducted experiments deliberately damaging 80–100% of OHCs in cochleae to release prestin from the OHCs. However, the concentration of prestin in the serum of these mice was not significantly different from those with undamaged OHCs or prestin-KO mice. These data raise the question of whether prestin proteins in the cochleae can pass the ‘blood-labyrinth barrier’ as previously thought.

OHCs are one of the most vulnerable components in the cochlea. Therefore, OHC proteins are considered a good choice as biomarkers for cochlear damage. There are several reasons why prestin is stands out as a potential biomarker. First, it was believed to be the unique protein only expressed in OHCs before it was found also expressed in the myocardium of hearts in 2021 [42]. Second, prestin is the most abundant membrane protein of OHCs. In this study, we found that the commercially available mouse prestin-ELISA kits cannot specifically detect prestin in mouse serum samples. It is unclear whether prestin-ELISA kits used for other animals and humans have the same specificity issue. The outstanding question is whether prestin is a potentially good biomarker for OHC stress/damage if ELISA’s specificity improves. It is generally believed that the prestin protein turnover rate is rather slow. The prestin-enriched lateral membrane renders membrane proteins virtually immobile, making it rare, if it is even possible, for OHCs to perform endocytosis/exocytosis from the lateral membrane [43]. There are few prestin proteins present in the cytoplasm after P16, as demonstrated by immunogold labeling electron microscopy [44]. Immunostaining for prestin can only detect prestin staining in the cytoplasm around the P5–P9 stage, presumably reflecting its production during development. There is no prestin staining at the basal membrane of matured OHCs, where endocytosis/exocytosis often occurs. These collected data suggest that prestin proteins are unlikely to be released into the bloodstream continually due to the proteomic recycling processes, as there is little free prestin present in OHCs’ cytoplasm or the basal membrane of OHCs to begin with. Based on recent Cryo-EM data, prestin forms dimers [4,5,6,7,8]. Each OHC is believed to have ~10^7^ 11 nm particles that are presumably made of prestin dimers, meaning that each OHC has ~2 × 10^7^ prestin molecules [45]. There are about 2000 OHCs in each mouse cochlea [46]. Thus, a total of 4 × 10^10^ (2000 × 2 × 10^7^) prestin molecules, equal to 6.6 × 10^−14^ moles of prestin in each cochlea (one mole has 6.02 × 10^23^ molecules). Assuming that all prestin molecules in both cochleae release from OHCs, all of them freely enter the bloodstream without any degradation or exclusion, and an adult mouse has about 2 mL of blood, we then would expect the concentration of prestin to be 6.6 × 10^−11^ M, which is not even past the detection limitation of the best prestin-ELISA kit, 7.8 pg/mL, i.e., 9 × 10^−10^ M (the MW of prestin is 81.4 KD). Therefore, despite being the most abundant membrane protein in OHCs, prestin molecules occupy only a minimal fragment of the bloodstream compared to the overall blood volume in animals or humans. Moreover, prestin is also expressed in cardiomyocytes. It remains to be investigated whether prestin from OHCs is a reliable and excellent serological biomarker for damaged OHCs.

## 4. Materials and Methods

### 4.1. Animal

All experimental procedures were conducted in accordance with the Guide for the Care and Use of Laboratory Animals by the NIH and were approved by Northwestern University’s Institutional Animal Care and Use Committee. Details on the generation and characterization of prestin-KO were described elsewhere [9]. To minimize age-related OHC loss commonly found in mice on the C57BL/6 background, adult WT and prestin-KO mice are on the FVB background, which was generated by backcrossing the original mouse model (129/C57BL/6 background) [9] to the FVB strain for more than ten generations, after which mice are then maintained for several years without refreshing the background. The FVB strain is known to have excellent high-frequency hearing well into adulthood [47]. Genotyping was outsourced to Transnetyx (Cordova, TN). Both males and females were tested in all experiments.

### 4.2. Cochlear Stress Treatment

HPβCD treatment: HPβCD (Sigma, H107) was dissolved in 0.9% NaCl injection USP (Baxter, Deerfield, IL). HPβCD and vehicle control (0.9% NaCl) solution were passed through a 0.22 μm syringe filter (Millipore) for sterilization before injecting into animals. Each adult WT and prestin-KO mouse was injected at a dose of 8000 mg per kg body weight of HPβCD or equivalent vehicle control subcutaneously, as described previously [35,36]. Both male and female mice were used. Cochleae and blood samples were collected at four hours (h), 1 day, or 4–11 days for immunofluorescence and ELISA.

Noise exposure: Eight male and female WT mice were exposed to bandlimited (8–16 k Hz) noise for 2 h at 110 dB SPL (re 20 µPa), which led to permanent hearing loss (PTS) [48]. Eight male and female mice that stayed in ambient conditions were used as the control group. Serums and cochlear samples were collected 2–4 h after noise exposure for immunofluorescence and ELISA.

### 4.3. Prestin and Hemoglobin Measurement

Serums were collected from clotted blood. Because mouse red blood cells were fragile and easily ruptured under physical pressure that caused hemolysis, we collected free-flow blood from depictured bodies without using any physical pressure or mechanical force. Briefly, after collecting the whole blood from decapitated bodies of euthanized animals, we allowed the blood to clot by leaving it undisturbed at room temperature for 30 min. Blood samples were centrifuged for 15 min at 1000× *g*. The samples were stored in a −80 °C refrigerator until the time of assay. Prestin concentration was measured using Mouse Prestin (SLC26A5) ELISA Kit from (1) MyBioSource Inc (MBS, MBS286559, San Diego, California), LSBio (LS-F65693-1, Shirley, MA), Abbexa (abx544297, Sugar Land, TX, USA). All the ELISAs were performed according to each manufacturer’s protocol. For most serum samples, 20 μL serum samples were mixed with 80 μL sample dilutant (1:5 dilution) provided by the manufacturers and placed in wells of prestin assay microplates. For some experiments performed using the MBS kit, we also used 1:10 dilution, aiming to decrease matrix effects as stated in the Results. The microplates were washed by an automatic microplate washer (Agilent BioTek 405 TS). The optical density in the wells of the ELISA microplate was measured at 450 nm and 540 or 570 nm using BioTek Synergy 2. Since hemolysis may influence ELISA data, hemoglobin’s absorbance at 414 nm was measured using a spectrophotometer, as reported previously [49].

### 4.4. Immunofluorescence

Mice were euthanized, and their cochleae were dissected out and fixed with 4% paraformaldehyde. After decalcification using 10% EDTA in PBS for 2–3 days, the tectorial membrane was removed, as we described previously [41]. A full-length cochlear surface preparation was then dissected out in PBS and cut into five pieces at specific locations for immune staining, following the instructions from the video published online by the Massachusetts Eye and Ear Infirmary (https://vimeo.com/144531710) (accessed on 3 November 2015). The whole mount samples were blocked at room temperature for one hour in a blocking solution: Tris-buffered saline (TBS) containing 5% goat serum and 0.2% saponin. Samples were incubated with the following primary antibodies at 4 °C overnight: Myosin 7A (1:50, Santa Cruz, sc-74516), N-terminal of mouse prestin (anti-N-mprestin), and C-terminal mouse prestin (anti-C-mprestin) rabbit antibodies (1:1000). As described in our previous publication, peptides MDHAEENEIPAETQRYYVER and TASLPQEDMEPNATPTTPEA were used to generate anti-N-mprestin and anti-C-mprestin, respectively [50,51]. The next day, samples were then washed in PBS and incubated with appropriate fluorophore-conjugated secondary antibodies for 2 h at room temperature. The secondary antibodies included goat anti-rabbit Alexa 488 at 1:500 (Thermo Fisher Scientific, Waltham, MA, USA) and goat anti-mouse IgG2b Alexa 647 at 1:500 (Thermo Fisher Scientific, Waltham, MA, USA). Alexa 546-conjugated phalloidin (Thermo Fisher Scientific, Waltham, MA, USA) was also used to stain actin as described before [35,36]. Stained cochlear sections were mounted onto slides using Dako fluorescent mounting medium (Agilent). Images were captured on a Keyence BZ-X800 microscope or Nikon A1R confocal microscope with Plan Fluor 10X, Plan Apo 20X, and a Plan Apo 60X oil objective (Nikon). Basilar membrane length was measured using ImageJ, and the numbers of remaining OHCs were determined. A mouse cochlear place-frequency map [52] was used to determine the corresponding frequencies. Based on the map and our measurements, the middle turn segment of our cochlear samples is in the frequency range between 19.1 and 36.5 kHz [41]. A cochleogram is usually used for determining the precise amount of OHC loss [53,54]. Since we did not find the cochleogram for FVB mice in the literature, we obtained the OHC density of the middle turn segment by counting OHCs from 5 cochlear samples, which was 394 ± 42 cells/mm. The percentage of OHC loss was then calculated based on this number.

### 4.5. Cochlear Lysate

Cochleae were dissected out from seven WT and seven prestin-KO mice separately. The collected cochleae were pooled together and homogenized using pellet pestles (Konts, 749520-000) in a lysis buffer containing TBS (pH 8.0) containing 2 mM DDM (n-Dodecyl-Beta-Maltoside), 2 mM DTT (Dithiothreitol), 1 mM PMSF (phenylmethylsulfonyl fluoride), and 1:50 protease inhibitor mixture (Sigma P8340). The samples were subjected to low-speed centrifugation (800× *g* for 5 min) to separate the bony structures. The supernatants were then sonicated, incubated on ice for 10 min, and centrifuged at 18,000× *g* for 10 min to remove cell fragments. The collected supernatant was stored at −80 °C for ELISA later.

### 4.6. Statistical Analysis

Statistical analysis was performed using GraphPad Prism 10 software. ELISA data were presented as the mean ± SD (standard deviations), and statistical analyses were performed using two-way ANOVA or mixed-effects model (REML), multiple unpaired *t*-tests, and one-way ANOVA, followed by Tukey’s multiple parisons test or unpaired Student’s *t*-test (two tails). *p* < 0.05 was considered to be statistically significant. The coefficient of variation (CV%) is the standard deviation divided by the mean (CV% = SD/mean). Sample data with intra-plate variation coefficients larger than 15% were unacceptable for analysis.

## 5. Conclusions

Prestin concentrations measured by ELISA kits are significantly affected by the quality of the collected serum. Prestin from OHCs is not a sensitive and reliable serological biomarker for detecting mouse cochlear damage using ELISA.

## Figures and Tables

**Figure 1 ijms-25-07285-f001:**
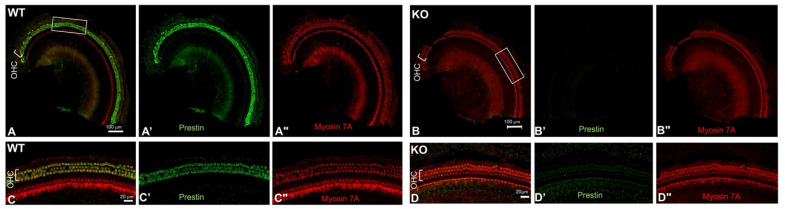
Representative confocal images show prestin expression patterns in whole-mount preparation of WT (**A**,**C**) and prestin-KO (**B**,**D**) cochleae. Antibodies include anti-N-mprestin (green) and anti-Myosin 7A (red). **C** and **D** are enlarged images of the boxed regions in (**A**,**B**), showing OHCs in both samples. (**A’**–**D’**) and (**A”**–**D”**) were individual channels for prestin and Myosin 7A staining. Bar: (**A**,**B**): 100 μm, (**C**,**D**): 20 μm. Number of samples (n): WT, n = 2; prestin-KO, n = 2.

**Figure 2 ijms-25-07285-f002:**
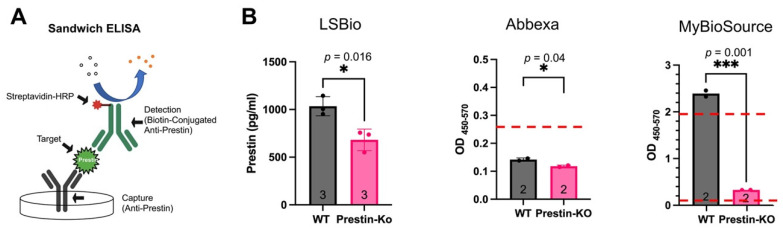
(A). The principle of a sandwich ELISA. (**B**). Prestin levels in cochlear homogenates of WT and prestin-KO mice were measured by three prestin-ELISA kits. The LSBio kit detected prestin in both WT and prestin-KO cochlear homogenates. The average prestin concentration: prestin-KO samples, 681 pg/mL; WT samples: 1034 pg/mL. The Abbexa kit was not sensitive enough to detect prestin because its OD_450–570_ numeric values for both WT and prestin-KO samples were below the detectable range, indicated by a red dashed line (78 pg/mL). OD_450–570_, measured by the MBS kit, has the biggest difference between WT and prestin-KO samples. Two red dashed lines indicate the detectable range for the MBS kit: 7.8–500 pg/mL. Each dot represents one technical repeat of pooled cochlear homogenates. Means ± SD were plotted, and significance was determined using a *t*-test. The number of samples and *p* were shown. *, *p* ≤ 0.05, ***, *p* ≤ 0.001.

**Figure 3 ijms-25-07285-f003:**
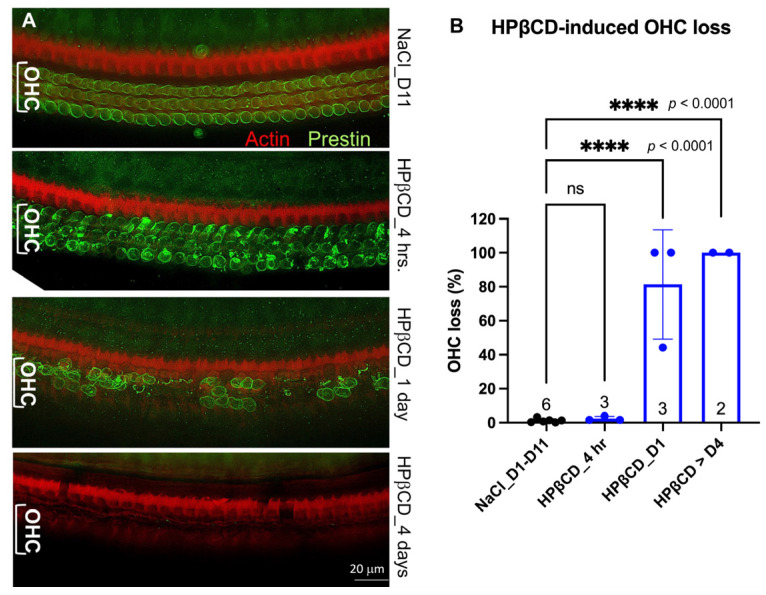
(**A**). Representative immunostaining images showing OHC loss in WT mice treated with HPβCD at different time points. Anti-N-mprestin (green) and phalloidin (red) were used to stain samples. (**B**). A histogram showing the average OHC loss in the middle turns of WT cochleae (frequency range: 19.1–36.5 kHz [41]). Each dot represents one animal sample. Means ± SD were plotted, and significance was determined using ordinary one-way ANOVA, followed by Tukey’s multiple comparisons test. The number of samples and *p* are also shown. D: day. hr: hour. Bar: 20 μm.****: *p* < 0.0001, ns: not significant.

**Figure 4 ijms-25-07285-f004:**
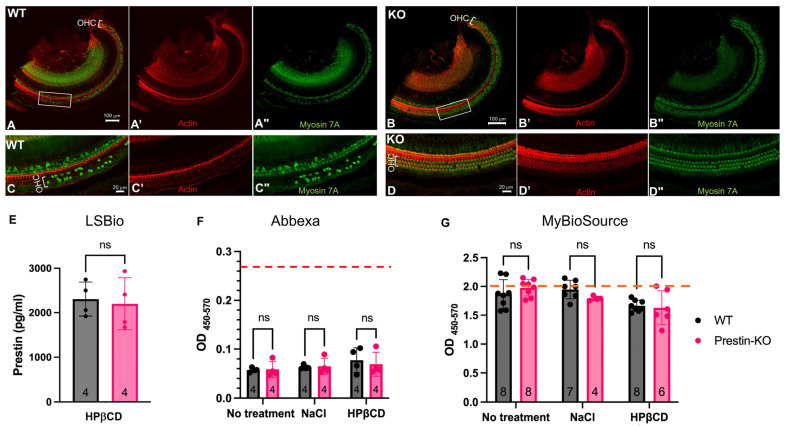
(**A**–**D**). Representative confocal images show OHC loss in mice one day after HPβCD injection. OHC loss in WT apical region (**A**,**C**) exceeds that in prestin-KO mice (**B**,**D**). Anti-Myosin 7A (green) and phalloidin (red) were used to stain IHCs and OHCs. (**C**,**D**) are enlarged images of the boxed regions in (**A**,**B**), showing OHCs in both samples. (**A’**–**D’**) and (**A”**–**D”**) were individual channels for actin and Myosin 7A staining. Bar: (**A**,**B**): 100 μm, (**C**,**D**): 20 μm. Number of samples (n): WT, n = 3; prestin-KO, n = 2. (**E**–**G**). Prestin levels were measured by three prestin-ELISA kits. WT and prestin-KO mice were injected with either HPβCD, NaCl, or no injection at all (no treatment). Serum samples were collected one day later for ELSIA using the LSBio kit (**E**), the Abbexa kit (**F**), and the MBS kit (**G**). Serum samples were diluted to 1:5 for LSBio and Abbexa, and 1:10 for MBS. The red dashed line shows the detectable low limit for the Abbexa kit (78 pg/mL) and the high limit (500 ng/mL) for MBS. The number of samples was also shown. Each dot represents one animal sample. Means ± SD were plotted, and significance was determined using a *t*-test for the LSbio ELISA (**E**), two-way ANOVA, and multiple unpaired *t*-tests for the Abbexa ELISA (**F**) and the MBS ELISA (**G**). ns: not significant.

**Figure 5 ijms-25-07285-f005:**
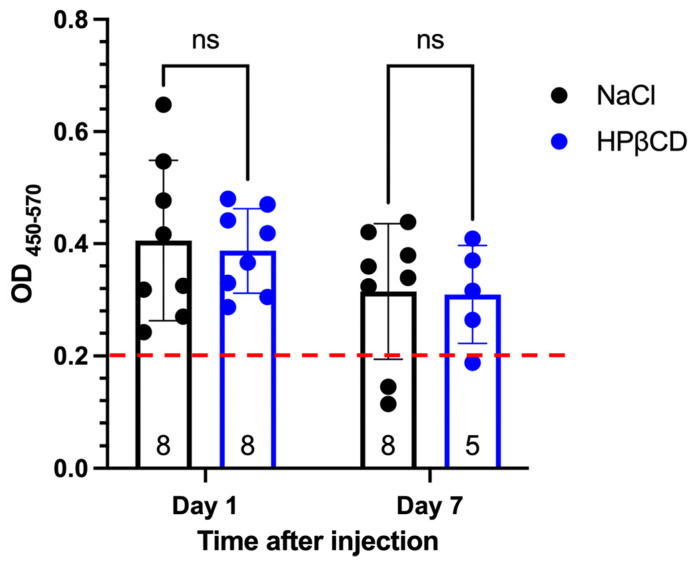
Prestin levels in serum samples of WT injected with HPβCD or 0.9% NaCl. The serums were collected at different time points after WT mice were injected with HPβCD or 0.9% NaCl. Prestin concentrations were measured using a mouse prestin-ELISA kit made by MBS. Each dot represents one animal sample. The number of samples was also shown. Means ± SD were plotted, and significance was determined using two-way ANOVA and multiple comparisons. The red dashed line shows the detectable low limit for the MBS kit (7.8 pg/mL). ns: not significant.

**Figure 6 ijms-25-07285-f006:**
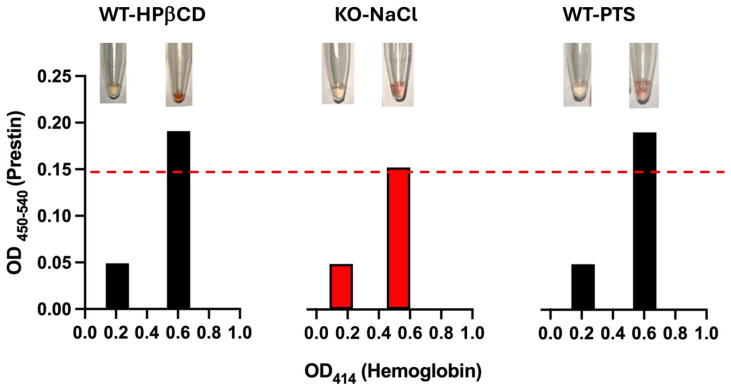
Prestin concentrations measured by the prestin-ELISA kit made by MBS are significantly altered by hemolysis. Hemolysis was determined by visual inspection (see pictures on the top) and absorbance at 414 nm (below). WT and prestin-KO mice were treated with HPβCD, NaCl, or noise (PTS). Two serum samples from each mouse—one with low and another one with high hemoglobin. Even though these two serum samples were collected from the same mouse at the same time, prestin concentrations were only detected in the samples with high hemoglobin, not in the ones with low hemoglobin. The red dashed line shows the detectable limit for the MBS kit (7.8 pg/mL).

## Data Availability

All data, including the original images, will be made available upon request to the corresponding author.

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
