# Peer review of "Verification of Outer Hair Cell Motor Protein, Prestin, as a Serological Biomarker for Mouse Cochlear Damage"

_ijms, 2024, doi:10.3390/ijms25137285_

Round 1

Reviewer 1 Report

Comments and Suggestions for Authors

Abstract
- The background, methodology, findings, and conclusions sections of the abstract could be divided into distinct sections for easier reading. This would aid readers in rapidly grasping the main ideas.  
- More details about the procedures and outcomes are needed, such as the quantity of mice utilized in each trial group.
- The abstract's conclusion that prestin is an unreliable biomarker should be softened because more data may be required to support such a strong claim. I would propose rewording the data to say that Prestin levels in this mouse model did not change in response to OHC injury.

Introduction:
- A solid foundation on prestin and its potential as a biomarker is given in the introduction.

- introduce the role of oxidative stress biomarkers in cochlear damage. cite doi:10.3390/life14040425.
- At the conclusion of the introductory section, it would be beneficial to provide a clearer statement of the study's precise goal or hypothesis.

Methods:
 - Further information is required on the number of animals used in each experiment group, the method used to assign them to groups, and the particulars of the procedures, such as the length and intensity of noise exposure.
- Were the trials repeated? How frequently were important experiments carried out? This data is critical for assessing the accuracy of the findings.
- Further details regarding the ELISA kits, such as the manufacturer and catalog numbers, are required.

Results:
- Although the results are presented coherently, they would be strengthened with more quantitative information.

- For instance, the WT and KO groups in Figure 1 should each have n numbers.  
 -It is recommended that Figures 2 and 4 use statistical analyses, such as t-tests, on the bar graphs to illustrate the importance of group differences.
- Displaying individual data points underneath the bar graphs in Figure 5 would improve the image. Since each group only has n=2, the means and SDs provide less information.

Discussion:
- The discussion would be improved by placing the results in the context of earlier research and providing plausible justifications for the model's lack of

- Overspending on the consequences of it being an unreliable biomarker in general could be warranted.  discuss the role of genetic mutations. cite doi:10.3390/biomedicines11061616.
changes in prestin with OHC injury when other models showed changes.

- The study's limitations, such as its small sample sizes, should be addressed.
The findings that Prestin did not show to be a reliable biomarker in this mouse model may need to be softened in light of the need for more research.

Comments on the Quality of English Language

any

Author Response

Thanks for the suggestions. We have modified the manuscript according to the suggestions from the Reviewers. The point-by-point responses to the reviewer’s comments (in blue) are listed below.

Abstract

- The background, methodology, findings, and conclusions sections of the abstract could be divided into distinct sections for easier reading. This would aid readers in rapidly grasping the main ideas.  

Thanks for the suggestion. We have changed the abstract into four sections: background, method, results, and conclusions.

- More details about the procedures and outcomes are needed, such as the quantity of mice utilized in each trial group.

The abstract has a 200-word limit, so we cannot fit all the information into it. We added more detailed information in the Method and Results sections, including animal numbers for each experiment.

- The abstract's conclusion that prestin is an unreliable biomarker should be softened because more data may be required to support such a strong claim. I would propose rewording the data to say that Prestin levels in this mouse model did not change in response to OHC injury.

We value the reviewer’s suggestion. We softened our conclusion by adding ‘using ELISA’ to define the condition of our results. We also changed the title to ‘Verification of Outer Hair Cell Motor Protein, Prestin, as a Serological Biomarker for Mouse Cochlear Damage.’ However, it's important to note that our conclusion is not based on a single experiment. Rather, it is supported by various experimental data and 25 years of accumulated knowledge about Prestin. This wealth of evidence underscores the validity of our findings. 

As stated in the introduction, numerous papers have utilized Prestin as a biomarker for cochlear damage through ELISA. However, none of these studies have employed a suitable negative control to validate the specificity of their ELISA. As the pioneering group in the discovery of Prestin, we and other research teams have developed dozens of anti-Prestin antibodies targeting different amino acids of Prestin. Despite our extensive efforts, we have not yet identified an antibody that exclusively detects Prestin. For instance, our best antibody can specifically identify Prestin in immunofluorescence in the cochlea, but it does not consistently do so in Western blot. This underscores the critical role of validated ELISA in detecting authentic Prestin in the serum. Our research is the first to utilize Prestin-KO as the negative control, a gold standard for confirming the specificity of ELISA. Our data showed that all commercial ELISA tests failed to detect Prestin specifically in the serum, further validating our approach. Furthermore, we explained the reasons why Prestin cannot serve as a sensitive biomarker, citing its large size and limited quantity. To substantiate our claims, we conducted experiments where we deliberately eliminated 80-100% of OHCs. This rigorous approach underscores the robustness of our conclusions, and we stand by them. 

Introduction:

- introduce the role of oxidative stress biomarkers in cochlear damage. cite doi:10.3390/life14040425.

Thanks for the suggestion. However, this reference is a review paper. Since there is ambiguity in this critical issue, and the review papers do not provide solid research data, we only cite the original research papers in this manuscript.

- At the conclusion of the introductory section, it would be beneficial to provide a clearer statement of the study's precise goal or hypothesis.

We stated in the last paragraph of the introduction that the goal of the study is to investigate whether Prestin is a serological biomarker for cochlear damage or stress. 

Methods:
 - Further information is required on the number of animals used in each experiment group, the method used to assign them to groups, and the particulars of the procedures, such as the length and intensity of noise exposure.

We listed the animal number for each experiment in the Result section (as shown in the text or figures) since it is in front of the Method section. We added animal numbers for noise exposure as suggested. We have also mentioned the intensity and length of noise exposure in the Method section for the noise exposure experiment.

- Were the trials repeated? How frequently were important experiments carried out? This data is critical for assessing the accuracy of the findings.

We performed ELISA ten times and each sample for ELISA was duplicated. As we mentioned in the methods, we set the rigorous criteria that intraplate CV%>15 was not accepted for analysis.

- Further details regarding the ELISA kits, such as the manufacturer and catalog numbers, are required.

We had listed the company’s name, catalog number, and locations for each ELISA kit in the ‘Methods’ section.

Results:
- Although the results are presented coherently, they would be strengthened with more quantitative information.

- For instance, the WT and KO groups in Figure 1 should each have n numbers.  

The animal number was stated in the text on page 3, line 89. As suggested, we also added n in the figure legend.

 -It is recommended that Figures 2 and 4 use statistical analyses, such as t-tests, on the bar graphs to illustrate the importance of group differences.

We used statistical analyses, with p and means ± SD in figure legends for Figures 2-5. The detailed statistical methods were also stated in the Methods section and figure legend.

- Displaying individual data points underneath the bar graphs in Figure 5 would improve the image. Since each group only has n=2, the means and SDs provide less information.

Indeed, we showed each individual data point for Figures 2 to 5. Each dot represented one animal sample for serum and a technical repeat for the cochlear homogenate.

Discussion:
- The discussion would be improved by placing the results in the context of earlier research and providing plausible justifications for the model's lack of

Thanks for the suggestion. A new paragraph was added to address our results in the context of data from other previous papers.

- Overspending on the consequences of it being an unreliable biomarker in general could be warranted.  discuss the role of genetic mutations. cite doi:10.3390/biomedicines11061616.
changes in prestin with OHC injury when other models showed changes.

This reference is a review paper. We only cite the original research papers. As we mentioned before, none of the previous papers tested the specificity of their ELISA, so the collected data was questionable.

- The study's limitations, such as its small sample sizes, should be addressed.
The findings that Prestin did not show to be a reliable biomarker in this mouse model may need to be softened in light of the need for more research.

We appreciate your suggestion. To make it more subtle, we have changed the title of our paper to ‘Verification of Outer Hair Cell Motor Protein, Prestin, as a Serological Biomarker for Mouse Cochlear Damage.’  We acknowledge that the number of animals used in our study is small. When we initially designed the experiments, we had a larger sample size. We excluded all samples with hemolysis.

Reviewer 2 Report

Comments and Suggestions for Authors

The authors challenge the established studies on using prestin levels to detect cochlear damage. This is a potentially import topic, however, it is hard to believe that the data lead their conclusion. Furthermore, it is argued in the introduction that other studies did not consider that prestin is also expressed in hearts. This article will contribute to the field significantly if this can be addressed.

The main issues are:

1.       The authors did not compare the methods used by other groups before making the conclusion. For example, they should try kit MBS109040.

2.       The manufacture’s manual should be read carefully before conducting experiments. For example, the detection range for MBS286559 is 15.6 – 1000 pg/ml, not 7.8-500 pg/ml. Some studies/kits require the samples to be centrifuged within 30 min from collection but all the blood samples in this study were left at room temperature for 30 min – how might this affect your study? Furthermore, it is stated in the manual that hemolysis will influence the results, therefore, the authors did not “discover” this and it is not necessary to prove this in section 12.5.

3.       There are not enough numbers for statistical analysis and the analysis method for each figure is not always stated. For example, in Fig 2, at least 3 data point should be provided. It should be stated the n numbers for both biological repeats and technical repeats.

4.       Even if their data was collected and analyzed correctly, the title does not fit the study. To be more precise, “prestin is not a reliable serological biomarker” should be changed to “ELISA cannot reliably detect prestin for cochlear damage”. This is because they only used ELISA to detect prestin while they should provide additional evidence to make sure that it is not the variability arise from ELISA that bias the analysis.

Other issues are:

1.       What are the epitopes for the prestin antibodies used in this study? How were the antibodies verified?

2.       Samples were collected separately for each kit. Have you tried to collect the samples all together and use the same sample pool for each kit to compare between kits?

3.       Please quantify OHC numbers instead of OHC loss. This is because no one can be sure how many cells are missing from a huge empty space with no labeling.

4.       Confliction between Fig 2 and 4. Fig 2 stating that prestin level from MBS kit is over the range while Fig 4 still shows the values as pg/ml?

Author Response

Thanks for the suggestions. We have modified the manuscript according to the suggestions from the Reviewers. The point-by-point responses to the reviewer’s comments (in blue) are listed below.

The authors challenge the established studies on using prestin levels to detect cochlear damage. This is a potentially import topic, however, it is hard to believe that the data lead their conclusion. Furthermore, it is argued in the introduction that other studies did not consider that prestin is also expressed in hearts. This article will contribute to the field significantly if this can be addressed.

As the group that discovered Prestin and has studied it for more than 25 years, we would be more than happy to use Prestin as the biomarker if the data supported it. This is also the original purpose of this study. Unfortunately, our data do not support the idea that Prestin is a reliable a serological biomarker for cochlear stress or damage.

We removed the sentence regarding Prestin expression in hearts from the Introduction section as suggested. 

  1. The authors did not compare the methods used by other groups before making the conclusion. For example, they should try kit MBS109040.

We tested all commercially available mouse Prestin kits that have reasonable sensitivity. We have several reasons for not using MBS109040:

  1. MBS109040 is designed for Cavy, but we used mice as our experimental model. There is no reason for us to use a Cavy kit to detect mouse serum.
  2. There are no KO models for Cavy, so we cannot verify the specificity of this ELISA kit.
  3. The detectable range of MBS109040 is 0.625 ng/ml - 20 ng/ml, while the ELISA kit designed for mice from the same company has a detectable range of 7.8-500 pg/ml, making it 80 times more sensitive than MBS109040.

We want to emphasize that some mouse Prestin ELISA kits in our study do detect Prestin signals in mouse serum (Figs 4-5), just like other groups have reported. The difference between our data and others is that we used prestin-KO to verify the specificity of Prestin-ELISA kits. All tested Prestin ELISA kits detect a similar amount of Prestin in the serums from both WT and prestin-KO mice (which should have no Prestin in any cells). These data strongly suggest that Prestin signals detected in the WT serum are false positives.

  1. The manufacture’s manual should be read carefully before conducting experiments. For example, the detection range for MBS286559 is 15.6 – 1000 pg/ml, not 7.8-500 pg/ml. Some studies/kits require the samples to be centrifuged within 30 min from collection but all the blood samples in this study were left at room temperature for 30 min – how might this affect your study? Furthermore, it is stated in the manual that hemolysis will influence the results, therefore, the authors did not “discover” this and it is not necessary to prove this in section 12.5.

We appreciate the reviewer’s careful observation and suggestion. The sensitivity for MBS286559 on their website is indeed 15.6 – 1000 pg/ml. We tried many different ways to verify our findings, including discussing with MBS to see if they could improve the sensitivity. Eventually, MBS customized its product for us, providing a range of 7.8-500 pg/ml. 

Regarding the condition for collecting serum, we tried different procedures, with waiting times ranging from 0 (immediate) to 2 hours or 4oC overnight, and we even tried plasma. The results were similar: there is no difference between WT and prestin-KO, regardless of the collection procedures.

Indeed, MBS mentioned the effect of hemolysis. Red cells do not express Prestin. In theory, broken red cells or any cells in the bloodstream should not affect the interaction between Prestin-antigen and their antibodies. However, the fact that we observed a dramatic difference in their OD reading indicates the non-specificity of Prestin-ELISA. We put the data in 2.5 to emphasize its importance.

  1. There are not enough numbers for statistical analysis and the analysis method for each figure is not always stated. For example, in Fig 2, at least 3 data point should be provided. It should be stated the n numbers for both biological repeats and technical repeats.

Thanks for the suggestions.  All ELISA samples were performed in duplicate. Serum sample sizes are biological repeats. Each dot represents one animal sample. Cochlear homogenates were technical repeats as they were pooled samples collected from the cochleae of 7 mice. Statistic methods were added to Figures 2-5 and in the Methods section.

  1. Even if their data was collected and analyzed correctly, the title does not fit the study. To be more precise, “prestin is not a reliable serological biomarker” should be changed to “ELISA cannot reliably detect prestin for cochlear damage”. This is because they only used ELISA to detect prestin while they should provide additional evidence to make sure that it is not the variability arise from ELISA that bias the analysis.

Thanks for the suggestions. We changed the title to ‘Verification of Outer Hair Cell Motor Protein, Prestin, as a Serological Biomarker for Mouse Cochlear Damage.’ We also added ‘using ELISA' in the conclusion.

Other issues are:

  1. What are the epitopes for the prestin antibodies used in this study? How were the antibodies verified?

For C-terminus anti-C-mPrestin: TASLPQEDMEPNATPTTPEA. See the reference:

Matsuda, K., et al., N-linked glycosylation sites of the motor protein prestin: effects on membrane targeting and electrophysiological function. J Neurochem, 2004. 89(4): p. 928-38.

For N-terminal anti-N-mPrestin:  MDHAEENEIPAETQRYYVER. See the reference:

Zheng, J., et al., The C-terminus of prestin influences nonlinear capacitance and plasma membrane targeting. J Cell Sci, 2005. 118(Pt 13): p. 2987-96.

Both antibodies have been verified by Presin-KO mice, a gold standard for confirming the specificity of an antibody.

  1. Samples were collected separately for each kit. Have you tried collecting the samples all together and using the same sample pool for each kit to compare kits?

We did not pool our serum for ELISA. For cochlear homogenates, we used the pool cochlear homogenates that were used for all ELISA kits with equal amount. 

  1. Please quantify OHC numbers instead of OHC loss. This is because no one can be sure how many cells are missing from a huge empty space with no labeling.

We understand the concern from the reviewer regarding the quantification of OHC loss. Most IHCs and pillar cells survived in HPßCD-treated mice and could be used to estimate the number of missing OHCs, given the known ratio of IHCs (pillar cells) to OHCs is 1:3. We believe using percentages is advantageous as it provides a clearer indication of the extent of hair cell loss in specific regions, offering a more topical understanding of the damage. We added the detail procedure regarding OHC loss (%) counting in Methods section.

  1. Confliction between Fig 2 and 4. Fig 2 stating that prestin level from MBS kit is over the range while Fig 4 still shows the values as pg/ml?

Figure 2 shows cochlear homogenates, while Figure 4 shows serum data, which are indistinguishable from the prestin-KO signals.

Reviewer 3 Report

Comments and Suggestions for Authors This manuscript shows that current commercial elisa kits for prestin concentrations cannot detect any differences in the concentration of prestin upon outer hair cell (OHC) damage. Despite the fact that authors provide statistically insignificant results, these results are important and may deserve publication since these are in contrast to several published studies Their animal model and their methodology seem appropriate as well as their hypothesis to explain their difference to the possitive results provided by previous studies. The abstract of the manuscript also provided all the necessary and important information on their research

Of course the fact that current commercial elisa failed to detect differences in prestin concentration upon OHC damage does not necessarily means that pepsin is useless as a biomarker. Thus i suggest authors modifying their tittle a little bit e.g change ..is not a reliable.. to ...may not a reliable.... Another minor change that I propose concerns the term "incredibly debilitating" in the first sentence. Maybe a simple "debilitating" is more appropriate in this scientific manuscript.

Comments on the Quality of English Language

minor changes have been suggested

Author Response

We appreciate the reviewer’s suggestion. The revised manuscript deletes the word “incredibly” and changes the title to ‘Verification of Outer Hair Cell Motor Protein, Prestin, as a Serological Biomarker for Mouse Cochlear Damage.'

Round 2

Reviewer 2 Report

Comments and Suggestions for Authors

The authors made an effort in making the narrative of the manuscript more accurate; however, it is difficult to be convinced when there are not enough n numbers for statistical analysis – the n number should enough to present a normal distribution. Please provide more n numbers to back up your conclusion.

Minor comments:

1. “MBS customized its product for us, providing a range of 7.8-500 pg/ml”. Please mention this in the methods.

2. As the detection of Prestin heavily depend on the antibodies used, please mention the details about the antibodies in the methods.

3. “The known ratio of IHCs (pillar cells) to OHCs is 1:3”. Calculation cannot be based on this assumption because sometimes there are 4 rows of OHCs. In addition, OHCs do not always align with IHCs, especially in the case that there are OHC loss. Please provide a calculation based on the remaining OHCs.

Author Response

Thanks for the suggestions. We have modified the manuscript according to the suggestions from the Reviewer 2. The point-by-point responses to the reviewer’s comments (in blue) are listed below.

The authors made an effort in making the narrative of the manuscript more accurate; however, it is difficult to be convinced when there are not enough n numbers for statistical analysis – the n number should enough to present a normal distribution. Please provide more n numbers to back up your conclusion.

We have included three new figures in the supplementary material and revised Figures 4 and 5 to include more animal numbers.

Minor comments:

  1. “MBS customized its product for us, providing a range of 7.8-500 pg/ml”. Please mention this in the methods.

We have purchased several Prestin ELISA kits from MBS, which were made using three different raw materials (specifically, different antibodies). The original ELISA kit had a detection range of 15.6-1000 pg/ml, as stated on their website. We asked MBS if they could change the detection range as our samples contain very low levels of Prestin. Consequently, MBS supplied us with new kits featuring a detection range of 7.8-500 pg/ml. In our new version, we have included data obtained from kits with different detection ranges.

  1. As the detection of Prestin heavily depend on the antibodies used, please mention the details about the antibodies in the methods.

In the new version, we included peptide sequences used to generate anti-N-mPrestin and anti-C-mPrestin, even though this information was published in the cited references. We do not know the detailed information regarding anti-Prestin antibodies used for Prestin ELISA kits made by different companies.

  1. “The known ratio of IHCs (pillar cells) to OHCs is 1:3”. Calculation cannot be based on this assumption because sometimes there are 4 rows of OHCs. In addition, OHCs do not always align with IHCs, especially in the case that there are OHC loss. Please provide a calculation based on the remaining OHCs.

We appreciate the comment of the reviewer. To obtain a precise number of the OHCs, we calculated the OHC density based on the OHC counting of 5 cochlear samples and got the number 394 cells/mm. The percentage loss was then calculated based on this number. We also described in more detail in the Materials and Methods 4.4.  "Based on the map and our measurements, the middle turn segment of our cochlear samples is in the frequency range between 19.1 and 36.5 kHz [41]. A cochleogram is usually used for determining the precise number of OHC loss [53, 54]. Since we didn’t find the cochleogram for FVB mice in the literature, we obtained the OHC density of the middle turn segment by counting OHCs from 5 cochlear samples, which was 394 ± 42 cells / mm. The percentage of OHC loss was then calculated based on this number."

Round 3

Reviewer 2 Report

Comments and Suggestions for Authors

No further comments.